# A Study on the Literacy Rate of Buddhist Monks in Dunhuang during the Late Tang, Five Dynasties, and Early Song Period

## Shaowei Wu

School of History and Culture, Shandong University, Jinan 250100, China; wushaowei5719@163.com

**Abstract:** Among the Dunhuang documents, when examining some of the monk signature lists, name list of monks copying scriptures and name list of monks chanting scriptures in monasteries, we can estimate a relatively accurate literacy rate of the Buddhist sangha. Generally speaking, the literacy rate of the sangha during the Guiyi Army 歸義軍 period (851–1036) was lower than that during the Tibetan occupation period (786–851). The reason for this change is closely related to each regime's Buddhist policy, the size and living situation of the sangha, and the Buddhist atmosphere. The decrease in the literacy rate of the sangha had great negative consequences, but when viewed under the context of the stay at home monks and the secularization of Buddhism, the number of literate monks had actually increased. They were more closely integrated with the secular society and their functions in the regional society were more pronounced. At the same time, the changes in the literacy rate of the monks in Dunhuang can also serve as an important reference for understanding the development of Buddhism in the Central China.

**Keywords:** Dunhuang; Buddhism; literacy rate; literate monks





## 1. Introduction

In the Middle Ages, the Buddhist sangha served an important function in regional societies. This influence is manifested through their religious authority on the one hand and their cultural knowledge on the other. For this reason, it is very important to understand their level of education—the literacy rate of the sangha.

Although there are many studies on literacy rates, research on the literacy rate of religious groups such as the sangha have always contained misunderstandings, and these groups were often rashly directly classified as literate groups (Mote 1972). The reason scholars have this impression may be related to the group's cultural activities. However, at least since the Northern and Southern Dynasties, there have been many illiterate monks and nuns in the sangha. *The Sūtra of the Dharma's Complete End (fa miejin jing* 法滅盡經), an apocryphal sutra dated to the end of the 5th century, revealed that many monks at that time "did not chant scriptures" 經不誦習 and "did not know characters and sentences".不識字句[1]. At the time that Wudi of Northern Zhou 北周武帝 (543–578) tried to exterminate Buddhism, he once required those who were illiterate to return to laity. Although the monk Tanji 曇積 strongly opposed this, he admitted that there were many monks who were "obtuse, and lacking a gift for reading. They study hard but have not learned a single character". 受性愚鈍, 於讀誦無緣; 習學至苦, 而不得一字. "They are not smart and they cannot read more than one phrase". 無聰力, 日誦不過一言. Some monks practicing asceticism were also incapable of "chanting".[2] It can be seen that the number of illiterate monks in the sangha in the late Northern and Southern Dynasties was obviously sizable enough to attract the attention of those in power. After the Tang and Song periods, illiterate monks still existed, and Cheng Minsheng even believed that as many as one third of the monks in the Song period might have illiterate.[3] Although he did not give the basis for his estimation, it should be true that there were a considerable number of illiterate monks in the Song period.

Both Western and Chinese academics have attempted to examine literacy rate in history, especially before modern times. However, due to the limitation of materials, these always end up as speculations at the end. As far as the research on literacy rate in ancient China is concerned, the materials used are mostly limited to handed down documents, so it is indeed difficult to discuss the problem of ancient literacy rate in depth. However, unearthed documents, especially the Dunhuang documents, have preserved a large number of source materials regarding contemporary life, including a bunch of materials that enable discussions on the literacy rate of Buddhist monks in the late Tang, Five Dynasties and early Song periods.

First, there is a wealth of information for this period on the total number of Buddhist monks in Dunhuang that is preserved in Dunhuang.[4] It should be noted that data on the size of the sangha is not available for each year, and in many cases there are discrepancies on dating between different data and sources on literacy. However, the ordination of monks were not held every year, and sometimes they would not be held for many years. Hao (1998) has observed that, like the Central China, where monks were not ordained annually, the "mahāyāna altars" (*fangdeng jietan* 方等戒坛) set up by the monks in Dunhuang to grant novice monks upasampada were also not set up every year, but only once every few years or even decades. In addition, during the Tibetan and Guiyi Army periods, there may also have been relatively strict government control over the sangha, and the total size of the sangha grew very slowly. For example, according to S. 2729 (1) "Report on Population Statistics Book of the Sangha Tribe of Shazhou under the Mi Jingbian in the Third Month of the Year of Dragon (788)" (辰年[788年]三月沙州僧尼部落米淨辯牒上算使勘牌子曆) (hereinafter referred to as "Report on Population Statistics Book") and P. 3060 "Record of Scripture Chanting by the Sangha on the Third Month of the Year 788" (788 *nian sanyue dunhuang sengtuan zhuanjing li* 788年三月敦煌僧團轉經歷). It can be seen that in the third month of 788, the number of male monks reached about 198. According to S. 5676V "Number of Monks and Nuns in the Monasteries of Shazhou Around the Year 825," (825 *nian zuoyou shazhou zhusi sengni shu* 825年左右沙洲諸寺僧尼數), we know that there were 218 male monks around 825 CE. In other words, in nearly forty years, the number of monks has increased by 20, and an average of one person has been added for each two years; if there were an average of twelve monasteries, each monastery increased by less than two people within forty years. Therefore, the size of the sangha would have remained roughly stable for a certain period of time and might have even decreased as the old monks passed away. As a result of this, the size of the sangha in a time period close to the one in question is also data that can be used as source material for literacy.

Second, during this period there are still a relatively large number of source materials available for an accurate estimate of the number of literate monks. An estimation of the number of literate people is one of the most important and difficult problems pertaining to ancient literacy. Due to the limitation of materials and methods, there is still a lack of discussion on the subject in academic circles. There are two premises to study this question, one of which is to determine a method for screening literate people. Research on literacy in recent times provided a method for screening literate texts, that is, the ability to "write" and "read" is always the key basis for confirming whether a person is literate. The second premise is that the basic social unit for research needs to be determined. Literacy rates require statistical treatment of a specific group rather than an individual. The social unit for the activity of the sangha is the monastery, so the smallest unit to discuss the literacy rate of the sangha is also that of the monastery. As far as the above conditions are concerned, there are three types of texts in the huge Dunhuang literature that can be used: the name list of monks copying scriptures organized by the monastery, the signature list left by all the monks in the monasteries when they participated in the decision-making of monastic affairs, and the scripture chanting name list for scripture chanting activities organized by the monastery.

The three types of source materials have different degrees of accuracy in calculating literacy rates. The "signature list" has been widely used in research on western literacy,

but in the view of modern literacy research, signature is just a "functional" expression of literacy, and cannot fully describe the literacy level of the writer. In contrast, the name list of monks copying scriptures and the scripture chanting name list are relatively unique source materials on the literacy rate of the sangha in the Middle Ages of China. Copying and chanting scriptures are literacy activities that last for a long time and dealing with more complex texts, so it can indicate that the participants are fully literate, which can better reflect the education level of the participants compared to signatures.[5]

Combining the aforementioned three types of source materials pertaining to literacy with other source materials, not only can we obtain a relatively accurate table for changes in the literacy rate of monks, but the factors affecting these changes in the literacy rate of monks in Dunhuang and the impact of these changes on different things can also be analyzed in depth. The implications of the changes in the number of literate monks for the local society can then be deeply explored. Moreover, although the form of Buddhism in Dunhuang during this period was different from the Buddhism of the Central China, it can still be used as a reference to understand Central Plains' Buddhism

## 2. List of Buddhist Scriptures and Signature List

The name list of monks copying scriptures is a list of people who are paid to copy Buddhist scriptures, and the signature list is a list of all the monks' signatures regarding some matters related to the collective interests of the monks in a monastery. These two types of source materials can reflect the writing ability of the monks, so they can be the basis for judging their competence in literacy.

First, let us look at the name list of monks copying scriptures. During the periods of Tibetan and Guiyi Army rule, the monasteries frequently organized Buddhist activities such as scripture chanting, and the damage and wearing on the scriptures were significant. As a result, activities for copying scriptures were frequently organized. Each time there was an organized activity, a name list of monks copying scriptures was made. Among the Dunhuang documents, there are a lot of source materials regarding copying scriptures, but only one of them has value for the estimation of literacy rate, namely the document S. 2711 "Name List of People Writing Scriptures in the Jinguangming Monastery and other Monasteries from the 810s to 820s" (810 *zhi* 820 *niandai jinguangming si deng si xiejing renming lu*, 810至820年代金光明寺等寺寫經人名錄).[6]

The beginning and end of this document is complete, and it records the name list of monks and some lay people who participated in copying scriptures at the Jinguangming Monastery. There are twenty two people who participated. They are: Jieran 戒然, Hong'en 弘恩, Rongzhao 榮照, Zhang Wuzhen 張悟真, Fazhen 法貞, Xianxian 賢賢, Sijia 寺加, Jinshu 金樞, Daozheng 道政, Fayuan 法緣, Liming 離名, Dong Fajian 董法建, Yizhen 義真, Huizhao 惠照, Qikong 霅空, Fachi 法持, Dao'an 道岸, Daoxiu 道秀, Chao'an 超岸, Tanhui 曇惠, Lisu 利俗, and Jingzhen 淨真. Also related to this document is P. 3205 "Report on Distribution of Work on Copying the Scriptures," (*chaojing fenpei li* 抄經分配曆) which records the specific tasks of copying the scriptures assigned to everyone on the basis of the document S. 2711.[7] The difference is that in P. 3205, there was an additional monk by the name of "Xiangyou" 像幽 from the Qianyuan Monastery 乾元寺. Copying scriptures is a fairly important source of income for the monks in Dunhuang, so on this occasion, the Jinguangming Monastery may have tried its best to call upon as many monks as possible to participate in copying scriptures. Under the circumstance that there was a shortage of people, some lay people and monks from other monasteries were also gathered. Therefore, I tend to think that the number of monks involved in writing scriptures here is relatively close to the number of literate monks in the Jinguangming Monastery at this time. The most recent data on the number of monks in the Jinguangming Monastery is found in S. 5676V "Number of Monks and Nuns in the Monasteries of Shazhou Around the Year 825". At that time, there were 26 monks in the Jinguangming Monastery. Therefore, the literacy rate of the monks in the Jinguangming Monastery at that time was roughly 84.6% (22/26).

Next, let us look at the signature list of the monks. The use of a signature list to discuss literacy rates has been used in the study of literacy rates in the West. For example, François and Jacques (1982) have used marriage registers to estimate changes in the literacy rate in France. They believe that in the 100 years from 1680 to 1780, the literacy rate of French adults increased from 40% to 70%. With the gradual deepening of the discussion on the literacy rate in recent times, the source materials used to analyze literacy rates have become more and more abundant, and the number of words people know has gradually become one of the central issues in the discussion of literacy. Therefore, whether signatures can be used for the study of literacy have given rise to some controversies.[8] Nevertheless, signatures can at least partly reflect the education level of the writer. Furthermore, as Liu (2017) said, writing Chinese characters is difficult to master without a long period of training; at the same time, the main writing utensil in ancient China was the soft brush, which required a higher level of control. Therefore, written Chinese characters (including signatures) can be used as evidence of literacy. Of course, some signatures are blunt and immature, indicating that the writer is not proficient in moving the brush, so such signatures should not simply be equated with literacy.

The signature lists used in this article are mainly those manuscripts that are collectively signed by monks in a monastery. The appearance of such manuscripts is related to the system by which important affairs of the sangha require internal co-determination. During the Sui and Tang periods, in the selection of the three directors (*san gang*) responsible for various affairs in the monastery, the state once stipulated that:

> Only those who use virtue and their abilities to transform their disciples, have the respect of both the clergy and the laity, and maintain the monastic rules can hold the position of the three directors. For all those who were nominated, their names need to be signed by themselves in a report and send to the officials.
>
> 凡任僧綱, 必須用德行能化徒眾, 道俗欽仰, 綱維法務者. 所舉徒眾, 皆連署牒官. [9]

During the periods of Tibetan and Guiyi Army rule, although the method of managing the monks in Dunhuang was adjusted, the system of co-determination within the monastery was still maintained. When there were matters in the monastery pertaining to the interests of all the monks, such as the election of monk officials and the settlement of income and expenditure, it was necessary to convene a general meeting of the whole monastery to make a resolution, and finally to collectively sign the report documents. This kind of source material can reveal at the same time the number of people who should have signed and the number of people who actually arrived. Therefore, this is one of the most ideal sources pertaining to statistics on literacy rates.

However, it should be noted that because speculations on writing abilities may be relatively subjective, when analyzing such materials, I do not intend to make a detailed classification, but simply divides them into two categories: those with writing ability and those with no writing ability. The ability to write is easy to understand; those without the ability to write are ones who cannot write names, which is mainly revealed in their clumsy brushwork. The reason for this determination is that if a person who does not practice writing often, his "brushwork will be very immature and rusty". (Liu 2017, pp. 107–8). On the other hand, if one's own signature is already very immature and rusty, then it can also be determined that the signer has almost no ability to write with the brush, and must be excluded in the estimation of literacy rates. Moreover, according to the following analysis of the documents P. 2049V (2) "Expenditure Record of the Head of Accountant on duty during a year (直岁 *zhisui*) of Jingtu Monastery for the Second Year of Changxing in the Later Tang (931)" (*houtang changxing er nian* [931] *jingtu si zhisui rupo li* 後唐長興二年 [931] 淨土寺直歲入破曆 and P. 2680V "Record on Distributing Scriptures Regarding Singing on the 'Mahāpranjāpāramitā Sutra' by the Jingtu Monastery in the Bingshen Year (936)," (*bingshen nian* [936] *jingtu si kai 'da bore' fujing li* 丙申年 (936) 淨土寺開〈大般若〉付經曆) it seems that it is more appropriate to exclude them from those who were literate. At the

same time, there is also the phenomenon of "empty space without writing" in this type of signature list, the reasons for which need to be analyzed in detail.

In the Dunhuang documents, I have found six such source materials,[10] which will be analyzed below:

(1) P. 3730 "The Report Pertaining to the Karmadāna Huaiying of the Jinguangming Monastery Inviting the Monk Huaiji to Fill the Position of Elder Along with the Verdict of Hongbian in the First Month of the Year of the Rooster". (*Tubo younian zhengyue jinguang-mingsi weina huaiying deng qing senghuaiji buchong shangzuo Zhuang bing Hongbian pan* 吐蕃酉年正月金光明寺維那懷 英等請僧淮濟補充上座等狀並洪辯判)

The beginning of this document is complete whereas the end is incomplete, and it recorded the Karmadāna Huaiying 懷英 of the Jinguangming Monastery inviting the monk Huaiji 淮濟 to fill the position of abbot in the first month of the Year of the Rooster under Tibetan rule. There was also a verdict of the head monk Hongbian 洪辯, and there was the signature of all the disciples agreeing to elect Huaiji and others at the end. According to the judgment of Tang and Lu (1990, p. 38), the "Year of the Rooster" of this document may be 829 or 841. However, the verdict in this document came from Hongbian, indicating that he was already the head monk of Dunhuang at that time. According to P. 4640 "Inscription of the head monk Wu" (*wu sengtong bei* 吳僧統碑) and the Tibetan inscription in Cave 365 of the Mogao Grottoes in Dunhuang, Wu Hongbian was appointed as chief instructor around 832. Therefore, the "Year of You" here should be the year 841.

Analysis of the writings of monks (see Table 1):

**Table 1.** Calligraphy of monks of Jinguangming Monastery in 841.

| | 1淮濟 Huaiji | 2智明 Zhiming | 3懷英 Huaiyin | 4善惠 Shanhui | 5智通 Zhitong | 6義藏 Yizang | 7神□ Shen □ | 8□□ | 9勝□ Sheng □ | 10法印 Fayin | 11談測 Tance |
|---|---|---|---|---|---|---|---|---|---|---|---|
| Picture | | | 懷英 | 善惠 | 智通 | 義藏 | 神□ | □□ | 勝□ | 法印 | 談測 |
| writing ability | yes | yes | yes | yes | yes | yes | yes | yes | yes | yes | yes |
| | 12玄秘 | 13法憲 Faxian | 14法顯 Faxian | 15智浪 Zhilang | 16靈秀 Lingxiu | 17法雲 Fayun | 18靈覺 Lingjue | 19法□ Fa □ | 20惠鋼 Huigang | 21迥秀 xiu | 22法象 Faxiang | 23□□ |
| | 玄秘 | 法憲 | 法顯 | 智浪 | 靈秀 | 法雲 | 靈覺 | 法□ | 惠鋼 | 迥秀 | 法象 | □□ |
| | yes | yes | yes | yes | yes | yes | yes | yes | yes | yes | yes | yes |

The end of this document is incomplete, so it is impossible to know for certain the number of monks in the Jinguangming Monastery at this time, and in the remaining part still show 23 people. Except for Huaiji and Zhiming, who were elected as monk officials and did not leave their signatures, the others left relatively nicely written signatures. However, since the former two were elected as the elder and abbot of the monastery, respectively, their ability to handle monastic affairs should be evident, and their writing skills should also be above the level of the typical monk.

The time period closest to this where we have data on the number of monks in the Jinguangming Monastery is in document S. 5676V "Number of Monks and Nuns in the Monasteries of Shazhou Around the Year 825". At that time, there were 26 monks in the Jinguangming Monastery. However, the two time periods are about 16 years apart, so the number 26 only has limited value as a reference. If the number 26 is used as the basis for estimation, then the minimum literacy rate of the Jinguangming Monastery in 841 should be above 88.5% (23/26).

(2) P. 3100 "The Report Pertaining to the monks of Lingtu Monastery inviting the master of Vinaya Shancai to Fill the Position of Abbot Along with the Verdict of Wuzhen in the the Second Year of Jingfu in the Tang Dynasty (893) (*Tang jingfu ernian lingtusi tuzhong gongying qing lvshi shancai chongsizhu zhuang ji dusengtong wuzhen pan* 唐景福二年 (893) 靈圖寺徒眾供英等請律師善才充寺主狀及都僧統悟真判)

The beginning and end of this document are both complete. It records that in 893 CE, the disciples offered the master of Vinaya, Shancai to be the abbot of the monastery, with the head monk Wuzhen making the judgment. Afterwards, there was a signature of agreement by everyone in the monastery to recommend Shancai. Although this document does not indicate the monastery's name, according to P. 3541 "*Epitaph to the portrait of Zhang Shancai*" (張善才邈真讚), we know that the abbot Shancai here is a monk of the Lingtu Monastery, so we know that the monastery in this document is the Lingtu Monastery.

A statistical analysis of the writings of the monks in the monastery is as follows (see Table 2):

**Table 2.** Calligraphy of monks of Lingtu Monastery in 893.

| | 1善才 Shancai | 2供英 Gongyin | 3龍夘 Longmou | 4 | 5 | 6慶□ Qin | 7 | 8 | 9靈龍 Linglong | 10張 Zhang | 11忠信 Xingzhong | 12慶□ Qin □ | 13道□ Dao □ | 14 | 15惠通 Huitong |
|---|---|---|---|---|---|---|---|---|---|---|---|---|---|---|---|
| Pictures |  |  |  | | |  | | |  |  |  |  |  | |  |
| writing ability | yes | yes | yes | | | yes | | | yes | no | no | yes | no | | no |

In this document, there are generally two monks' signatures in one column, but only one monk's signature was found in the last column, indicating that nothing is missing in the signature portion; the spaces below some of the "disciples" are left empty. It should be noted that these reserved spaces were not always at the end, implying that when the disciples were listed at that time, a certain format was followed (perhaps based on seniority or the status of the monks in the monastery). Even if some of the monks did not sign, others could not easily sign in his space. This also shows that this document should have listed all the disciples who needed to sign at that time; that is to say, at that time, there were probably 15 monks in the Lingtu Monastery, and 5 of them left empty spaces and did not write anything. According to the document that is 12 years later than this document, S. 2575 (3) "The Report Pertaining to the monks of Lingtu Monastery inviting Daxing to Fill the Position of Abbot in the August of the 5th year of Tianfu (905)" (*tianfu wunian* [905] *bayue lingtu si tuzhong qin daxing chong sizhu zhuang* 天復五年 (905) 八月靈圖寺徒眾請大行充寺主狀), which will be discussed below, it can be seen that in 905, the five monks "Daxing 大行, Yishen 義深, Lingjun 靈俊, Linglong 靈龍, and Zhengxin 政信" are not seen in this document, and Daxing is the new abbot, whereas Yishen is the elder of the monastery. They are both respected senior monks and not likely to have been new entrants from 893 to 905, so they are probably the ones with the five empty spots in this document. We also know from S. 2575(3) that with the exception of "Zhengxin," although the other four people varies in their writing abilities, they all had the ability to write their own names.

The writing style of people such as "Zhang 張, Zhongxin 忠信, Dao □ 道□, Huitong 惠通" in this document is very crude, which means their writing abilities were relatively low, maybe even at the level of beginners. The fact that even their own names were written so awkwardly indicates that their literacy is almost nonexistent. Therefore, the literacy rate of the monks in the Lingtu Monastery at that time was 66.7% (10/15).

(3) S. 2575 (3) "The Report Pertaining to the monks of Lingtu Monastery inviting Daxing to Fill the Position of Abbot in the August of the 5th year of Tianfu (905)" (*tianfu wunian* [905] *bayue lingtu si tuzhong qin daxing chong sizhu zhuang* 天復五年 (905) 八月靈圖寺徒衆請大行充寺主狀)

The beginning of this document is complete whereas the end is incomplete. It records that in the fifth year of Tianfu (905), the disciples of Lingtu Monastery asked Daxing to be the head of the monastery. They send in their report to the director of monks; the back of the report has the joint signatures of the monks who agreed to nominate D.

A statistical analysis of the writing of the monks in the monastery is as follows (see Table 3):

**Table 3.** Calligraphy of monks of Lingtu Monastery in 905.

|  | 1大行<br>**Daxing** | 2義深<br>**Yishen** | 3靈俊<br>**Lingjun** | **4** | 5政信<br>**Zhengxin** | 6惠<br>**Hui** | **7** | 8靈龍<br>**Linglong** |
|---|---|---|---|---|---|---|---|---|
| picture |  | 義深 | 靈俊 |  | 政信 | 惠 |  | 靈龍 |
| writing ability | yes | yes | yes |  | no | no |  | no |

The text of the report is partially incomplete, but there are no missing parts pertaining to the signatures of the disciples, so we know that in 905, there may have been eight monks in the Lingtu Monastery. In the report, there are two monks who left empty spaces and did not write. According to the above analysis, we know that those who left empty spaces but not writing do not necessarily mean that they did not have the ability to write. There is only one monk (Linglong 靈龍) from this document that is also seen in P. 3100. If the "Huitong" in P. 3100 is the "Hui" found in this document, there are still only two monks. In 893, there were still ten people who were literate in the Lingtu Monastery, so it is unlikely that only the two of them were left. Therefore, the disciples who left empty space with no writing in this document are probably also found in P. 3100 and had the ability to write. In the signatures from this report, Zheng Xin's handwriting is rather crude, implying that his knowledge is quite limited. Similar to this is the monk "Hui". The fact that they write their own names so crudely shows that their literacy level was very low. Excluding the "Zhengxin" and "Hui", the ratio of those who are literate would be 75% (6/8).

(4) P. 2049V (1) "Expenditure Record of the Head of the Jingtu Monastery for the Third Year of Tongguan in the Later Tang (925)"(*houtang tongguang sannian jingtusi rupo li* 後唐同光三年 (925) 淨土寺入破曆)

The beginning and end of P. 2049V are complete, and there are two copies of the expenditure record, namely P. 2049V (1) and P. 2049V (2) "Expenditure Record of the Jingtu Monastery for the Second Year of Changxing in the Later Tang (931)".(*houtang changing ernian jingtusi rupo li* 後唐長興二年 (931) 淨土寺入破曆) Both Expenditure Record keep the signature list.

A statistical analysis of the writings of the monks based on P.2049V (1) is as follows (see Table 4):

**Table 4.** Calligraphy of monks of Jingtu Monastery in 925.

| | 1保護 Baohu | 2淨Jing | 3保保 Baobao | 4道會 Daohui | 5寶Bao | 6道Dao | 7應Ying | 8法深 Fashen | 9願達 Yuanda | 10保達 Baoda |
|---|---|---|---|---|---|---|---|---|---|---|
| picture |  |  |  |  |  |  |  |  |  |  |
| writing ability | yes | yes | no | yes | no | no | no | yes | yes | no |
| | 11因會 Yinhui | 12功□Gong□ | 13願真 Yuanzhen | 14淨戒 Jingjie | 15古Gu | 16 | 17 | 18願濟 Yuanji | 19紹宗 Shaozong | 20 | 21 |
| |  |  |  |  |  | | |  |  | | |
| | no | no | yes | yes | no | | | yes | yes | | |

Therefore, according to the statistics of this document, we know that in 925, there were probably 21 monks in the Jingtu Monastery, of which four left empty spaces and did not sign, and eight had crude and clumsy handwriting. If these eight people were excluded from the literate group, then at this time, the literacy rate of monks in the Jingtu Monastery is 61.9% (13/21). In fact, according to the following analysis of P. 2680V "Record on Distributing Scriptures Regarding Singing on the '*Mahāpranjāpāramitā Sutra*' by the Jingtu Monastery in the Bingshen Year (936)," these people were not competent enough for the activities of reading the scriptures, that is, their literacy is very limited, and they should be excluded from being considered as part of the literate group.

(5) P. 2049V (2) "Expenditure Record of the Jingtu Monastery for the Second Year of Changxing in the Later Tang (931)".(*houtang changing ernian jingtusi rupo li* 後唐長興二年 (931) 淨土寺入破曆)

The statistical analysis of the signatures of the monks in the monastery mentioned in P.2049V (2) is as follows (see Table 5):

**Table 5.** Calligraphy of monks of Jingtu Monastery in 931.

| | 1願達 Yuanda | 2 | 3 | 4 | 5 | 6道會 Daohui | 7寶Bao | 8道Dao | 9法深 Fashen | 10應Ying | 11保達 Baoda | 12因會 Yinhui |
|---|---|---|---|---|---|---|---|---|---|---|---|---|
| picture |  |  |  |  | |  |  |  |  |  |  |  |
| writing ability | yes | no | no | no | | yes | yes | no | yes | no | no | no |
| | 13□□ | 14願真 Yuanzhen | 15淨戒 Jingjie | 16保護 Baohu | 17 | 18 | 19 | 20 | 21 | 22 | 23 | 24願濟 Yuanji | 25紹宗 Shaozong |
| |  |  |  |  |  | | | | | | |  |  |
| | yes | yes | yes | yes | no | | | | | | | yes | yes |

According to the statistics of this document, in 931, there may have been twenty five monks in the Jingtu Monastery, an increase of four compared with the year 925. Among them, there are fifteen people who also appeared in the previous document. There are seven people who left spaces and did not sign this document, but the names "Shanhui, Baohui, Baosheng, Jingsheng, Yuansheng, and Guangjin" contained in P. 2680V (8) that will be discussed below are not seen in this document. Furthermore, they are all monks who

are capable of chanting scriptures for a long time and have a high level of literacy. They are unlikely to be novice monks who were newly ordained during the five years from 931 to 936, so they are probably the six people who have not signed this document. In addition, the monk "Gong □" in P. 2049V (1) who had very poor hand writing is also not found in this document, and may also be one of those who did not sign. Including him, there are nine people whose hand writings are crude at this time. Therefore, overall, the literacy rate of the monks in Jingtu Monastery at this time was 64% (16/25).

(6) 羽52 "Expenditure Record of the Dayun Monastery in the Second Month of the Third Year of Yongxi of the Song Dynasty(986)"(*song yongxi sannian eryue dayunsi rupoli* 宋雍熙三年 (986) 二月大雲寺入破曆)

This document is incomplete at the beginning and complete at the end. At the end of the fascicle, there are the signatures of the disciples of the monastery.

The statistical analysis of the writings of the disciples in the monastery is as follows (see Table 6):

**Table 6.** Calligraphy of monks of Dayun Monastery in 986.

| | 1定惠 **Dinghui** | **2** | **3** | **4** | **5** | **6** | **7祥Xiang** | **8右You** | **9** | **10** | **11** |
|---|---|---|---|---|---|---|---|---|---|---|---|
| Picture | | | | | | | | | | | |
| Writing Ability | yes | yes | no | no | yes | yes | no | no | no | | yes |
| | **12** | **13** | **14** | **15** | **16** | **17** | **18定惠 Dinghui** | **19僧正 Sengzheng** | **20惠Hui** | **21僧正 Sengzheng** | **22護戒 Hujie** |
| | | | | | | | | | | | |
| | yes | yes | yes | yes | yes | | yes | yes | yes | yes | yes |

According to the statistics of this document, in 985 CE, there were probably 22 monks in the Dayun Monastery, of which 2 left empty spaces and did not sign, and 5 had crude and clumsy handwriting. Excluding them, the literacy rate was 63.6% (14/22). It is worth noting that at least 14 people in the signature list of this document used *huaya* 畫押 which is a kind of non-literal signature symbol instead of formal characters, accounting for as high as 70% (14/20) of the monks who signed. Although some economic documents in Dunhuang often uses *huaya* signature rather than standard characters, this signature list is indeed very unique compared to the previous ones.

## 3. The Scripture Chanting Name List of the Monastic Members

The monastery's scripture chanting name list records the list of monks who participated in the activity of reading scripture with rhythm. Scripture chanting is an activity of chanting and praying for blessings, and it is one of the most important and frequently held Buddhist activities of the sangha. When participating in chanting scriptures, especially when chanting major works such as the 600 fascicles of the *Mahāpranjāpāramitā Sutra*, reading is the most common method, which requires monks to be literate. The scripture chanting in the Dunhuang region includes the scripture chanting of all the monks of the Dunhuang sangha, and also the scripture chanting involving mainly one monastery along with some monks from other monasteries. There are also cases of the chanting of scriptures within a single monastery, and also cases of several people or even one person participating. Among the scripture chanting name list, there are only five documents that have value for estimating the literacy rate.[11] The analysis is as follows:

(1) S.10967 "Record of Scripture Chanting among Monks in the Lingtu Monastery and Other Monasteries Around the Year 789" (*789 nian qianhou lingtusi deng si sengzhong zhuanjing li* 789年前後靈圖寺等寺僧眾轉經歷)

The beginning and the end of this document are both complete, and it is the scripture chanting record of all of the monks of Dunhuang. According to this document, there were thirty four people who participated in scripture chanting by the names of Fayou 法幽, Biankong 辨空, Jinding 金頂, Weiji 維濟, Jieying 戒盈, Tanbian 曇辯, Baoyi 寶意, Rijun 日俊, Jintian 金田, Zhiyin 志殷, Abbot Ji (寂寺主), Elder Zhan (湛上座), Jianxin 堅信, Fajun 法濬, Xiuhui 修惠, Rigan 日幹, Fa Xing 法行, Zhi Cheng 至澄, Zheng Qin 正勤, Fa Quan 法詮, Hui Guang 惠光, Zhi En 智恩, Guang Zhao 光照, Wen Hui 文惠, Xiang Hai 像海, Esoteric Master (*mi fashi* 秘法), Jingu 金鼓, Huai'en 懷恩, Tanyin 曇隱, Jiao faxing 交法行, Liming 離名, and Fazhou 法舟. Other than "Zhi'en" who can be found in P. 3060 "Record of Scripture Chanting by the Sangha on the Third Month of the Year 788" and will be discussed below, the other monks' names are all found in S. 2729 (1). This indicates that this documents are also probably dated to around the year 789. Furthermore, except for the "Esoteric master" whom we cannot determine with certainty whether he is from the Qianyuan Monastery 乾元寺 or the Bao'en Monastery 報恩寺, we can determine the monastic affiliation of all the other monks. Among them, Lingtu Monastery had 13 monks, while the other monasteries had at most only 3 monks. It can be seen that this activity of chanting scriptures was organized with the Lingtu Monastery as the center. Among the monasteries, since only the Lingtu Monastery had a sizable number of monks, so when estimating the literacy rate, the estimates for the Lingtu Monastery are more valuable. According to P. 3060, it can be seen that after the third month of 788, the Dunhuang sangha increased by 79 monks and nuns, and the Bao'en Monastery and other monasteries increased the number of monks by 61. Since we cannot determine the increase in the number of monks in each monastery, it is not possible to discuss the literacy rate of the Lingtu Monastery and other monasteries at this time based on S.10967; however, the document can be used to discuss the literacy rate of the Lingtu Monastery in the third month of 788. According to this document, we know that the literate monks in the Lingtu Monastery are (the corresponding monk names in S. 2729 in parentheses):

Fayou (Jing Fayou 法幽), Abbot Song (Song Zhiji 宋志寂), Elder Zhan (Cao Zhizhan 曹志湛), Zhengqin (Song Zhengqin 宋正勤), Faquan (Chen Faquan 陳法詮), Guangzhao (Zhang Guangzhao 張光照), Zhicheng (Suo Zhicheng 索志澄), Tanbian (Ma Tanbian 馬曇辯), Wenhui (Zhang Wenhui 張文惠), Liming (Cao Liming 曹離名), Guangzhao (Zhang Guangzhao 張光照), Jingu (Li Jingu 李金鼓), Zhiyin (Zhang Zhiyin 張志殷).

According to S. 2729 (1), it can be seen that in the third month of 788, there were 17 monks in the Lingtu Monastery, so the literacy rate of the monastery at that time was 76.5% (13/17).

(2) P.3060 "Record of Scripture Chanting by the Sangha on the Third Month of the Year 788" (*788 nian sanyue Dunhuang sengtuan zhuanjing li* 788年三月敦煌僧團轉經歷)

The beginning and end of this document are both complete. It records a scripture chanting activity that involved the entire sangha. The end of the document recorded that there were "one hundred and three monks and seventy seven nuns". This meant that the total number of people who read scriptures on this occasion is 180.

Among the 180 people who chanted scriptures on this occasion, 100 (44 male monks, 56 nuns) were found in S. 2729 (1) and 80 people (59 male monks, 21 nuns) were not found in S. 2729 (1), indicating that this scripture chanting should have occurred shortly after the third month of 788. In the third month of 788, the size of the Dunhuang sangha was 310, which means that in just a few months, the size of the sangha increased by 25.8% (80/310). They were probably the result of the Tibetan regime bringing monks from other conquered areas and concentrating them in Dunhuang. Furthermore, according to S. 2729 (1), it can be seen that the "Fada" 法達 who participated in the scripture chanting died on the first day of the fourth month of the year of the dragon (788). "Jingfa 淨法, Chujing 處淨, and

Zhiming 智明" died in the first month and the eleventh month of the year of Horse (790), so this scripture chanting should have taken place in the third month of 788, and after the creation of the population statistics book to count the monks. S. 2729 (1) shows that there were 139 monks and 171 nuns in Dunhuang when the population statistics book was made. Therefore, during the third month, the number of male monks was about 200 (61 + 139), and the number of nuns was about 189 (18 + 171). In sum, the ratio of male monks chanting scriptures was 51.5% (103/200), and the ratio of nuns chanting scriptures was 40.7% (77/189).

However, comparing this document to S. 2729 (1), we can also find that not all the monks in the latter participated in this scripture chanting. The participation rate of male monks in S. 2729 (1) is 31.7% (44/139). In regard to the specifics of individual monasteries: Longxing Monastery 龍興寺 21.4% (6/28), JinguangMing Monastery 43.8% (7/16), Dayun Monastery 43.8% (7/16), Lingtu Monastery 35.3% (6/17), Bao'en Monastery 22.2% (2/9), Yong'an Monastery 9.1% (1/11), Liantai Monastery 蓮臺寺 27.3% (3/11), Qianyuan Monastery 31.6% (6/19), Kaiyuan Monastery 30.7% (4/13); the participation ratio of nuns is 32.7% (56/171), and the literacy rate figures for each convent is as follows: Lingxiu Monastery 靈修寺 35.8% (24/67), Puguang Monastery 普光寺 31.9% (15/47), and Dacheng Monastery 大乘寺 38.6% (17/44). According to the analysis of S. 10967, there were as many as 13 people in the Lingtu Monastery at that time, however, on this occasion, only 6 monks appeared. Therefore, the list here obviously did not include all the literate monks in each monastery; Dayun Monastery and the Jinguang Monastery had the highest rates, but even these monasteries only had 43.8%. Compared with the monasteries for monks, the number of new nuns added to convents was relatively less, with an average of 6 people per monastery; there were also many people involved in scripture chanting in each monastery, so the literacy level for document S.10967 may be closer to the literacy level of nuns in the third month of 788.

(3) P.3947 "Record of Thirty One Monks Chanting Scriptures at Longxing Monastery in the Year 831 or 843" (831 *huo* 843 *nian longxing si ying zhuanjing sanshi yi ren fenfan lu* 831或843年龍興寺應轉經冊一人分番錄)

The beginning and end of this document are both complete, and it is recorded by Cai Yin蔡殷, the minister of Longxing Monastery. The monks who participated in the scripture chanting activity are: Abbot Li 李寺主, Abbot Zhai 翟寺主, Du *Falü* 杜法律, Guizhen 歸真, Zhihai 智海, Changxing 常性, Huigui 惠歸, Zhenyi 真一, Faqing 法清, Judge Yong 顒判官, Boming 伯明, Shaojian 紹見, Fazhu 法住, Shengui 神歸, Lingzhao 靈照, Ling'e 靈尊, Guangzan 光讚, Huihai 惠海, Fuzhi 福智, Fazang 法藏 Bajie Duan 八戒段 (?), Abbot Duan 段寺主, Zhenmin 貞湊, Yingxiu 英秀, Fazhi 法智, Guo Fatong 郭法通, Huisu 惠素 (?), Huichang 惠常, Zhizhen 志真, Bi'an 彼岸, Haiyin 海印, Weiying 惟英, Farong 法榮, Fali 法利, Guangxi 光璨, Shenzang 神藏, Lingying 靈應, Daozhen 道珍, Lingxiu 靈秀, Fayin 法印, and Judge Deng 鄧判官, 41 monks total. The time of this scripture chanting is in a certain "year of Pig".

None of the monks in this document were seen in S. 2729 (1) of 788, so we know that there is a long gap in time between the "year of Pig" in this document and the year 788. At the same time, the monks in this document are often found in other documents. For example, Abbot Li, Du *Falü*, Guizhen, Zhihai, Faqing, Shaojian, Boming, Lingzhao, Huihai, Fazang, Zhenju, Fatong, Bi'an, Haiyin Weiying, Farong, Guangxi, Daozhen, Lingxiu, Fayin," twenty total, were also seen in P.t. 1261V "Record on Distributing the payment for performing ritual activities to Monks under the Tibetan Period" (*tufan shiqi sengren fenpei zhachen li* 吐蕃時期僧人分配齋傔曆) (Zheng 2001, p. 129). Therefore, the "the year of Pig" in this document may be the year 843, 831, or 819. Regarding the size of the sangha of the Longxing Monastery, among the source materials we have, the ones with the closest time period are S. 5676V "Number of Monks and Nuns in the Monasteries of Shazhou Around the Year 825," and S. 2614V "Monks and Nuns of the Monasteries in Shazhou at the End of the Ninth Century and the Beginning of the Tenth Century" (*jiu shiji mo shi shijie chu shazhou zhusi sengni mingbu* 九世紀末十世紀初沙州諸寺僧尼名簿). The number of people in

the Longxing Monastery in the former is 23, which is much lower than the 41 people in this document, so the date of this document is unlikely to be the year 819. The number of people of the Longxing Monastery in the latter is 50, although the date is a little later. Considering that the number of the monks in the Tibetan and Guiyi Army periods generally showed a trend of continuous growth, so in the "the year of Pig" when the scripture chanting took place, the size of the sangha at the Longxing Monastery is unlikely to have exceeded 50. Therefore, if 50 is used as the base of calculation, then the literacy rate of the Longxing Monastery at this time is roughly 82% (41/50).

(4) S. 11352 "Board Pertaining to Scripture Chanting of the Anguo Convent from the End of the Ninth Century to the Beginning of the Tenth Century" (*shi shiji mo shi shiji chu anguo nisi zhuanjing bang* 九世紀末十世紀初安國尼寺轉經牓)

The beginning and end portions of this document are both complete, and the content pertains to scripture chanting at a certain monastery. The text is as follows:

1. ☐ 　心 ｜ 政心　延惠
2. ☐ 　政惠　妙定　戒乘
3. 忍 ｜ 堅藏　真惠　圓智　妙林　真如
4. ☐嚴　如意　如吾　無性
5. ☐　政信　朱 (殊) 勝過 (果)　真頂　真行　能修　照
6. 心 ｜ 妙嚴　能寂　政思
7. 第二番: 慈藏　真願　妙行　濟實　朱勝智　如惠
8. 明會　堅忍
9. 右件, 國家轉經福田攘災, ☐宜宿
10. 不得寬☐, 如有故違, 必照重
11. 罰。☐☐謗 (牓) 示。
12. 今月廿三日法律道哲

1. ☐ 　Xin 心 ｜ 政心 Zhengxin 延惠 Yanhui
2. ☐ 　Zhenghui 政惠　Miaoding 妙定　Jiecheng 戒乘
3. Ren 忍 ｜ Jianzang 堅藏　Zhenhui 真惠　Yuanzhi 圓智　Miaolin 妙林　Zhenru 真如
4. ☐ Yan 严 Ruyi 如意 Ruwu 如吾 Wuxing 无性
5. ☐　Zhengxin　Shushengguo　Zhending　Zhenxing　Nengxiu　Zhaoxin
6. Xin　Miaoyan　Nengji　Zhengsi
7. Second Part: Cizang　Zhenyuan　Miaoxing　Jishi　Shushengzhi　Ruhui
8. Minghui　Jianren
9. On the right, the state organizes scripture chanting for fortune to avoid disasters, everyone should strict adherence to the requirements.
10. No leniency is allowed. If there is any violation, there will be serious
11. penalty. ☐ ☐ displays on board.
12. On the 23rd of this month, the *Falü* Daozhe

According to the recorded text, each line is about 6–7 people when complete, so the number of people may be around 40. The nuns in this document are all found in the Anguo Convent in S. 2614V. At that time, there were 100 nuns in the Anguo Convent, 23 śikṣamāṇās and 16 śrāmaṇeri, totaling 139 people. In this way, the ratio of those participating in this scripture chanting is roughly: 28.8% (40/139).

(5) P. 2680V "Record on Distributing Scriptures Regarding Singing on the 'Mahāpranjāpāramitā Sutra' by the Jingtu Monastery in the Bingshen Year (936)," (*bingshen nian* [936] *jingtu si kai 'da bore' fujing li* 丙申年 (936) 淨土寺開〈大般若〉付經曆).

This document is complete at the beginning and incomplete at the end. It was a record for distributing scriptures pertaining to read the *Mahāpranjāpāramitā Sutra* by a certain monastery in the Bingshen year. It involves monks: the governor of sangha (*Du Sengzheng* 都僧正), monastery governor Wu (*Wu Sengzheng* 吳僧正), Shanhui 善惠, Jingjie 淨

戒, Yuanzhen 願真, Baoda 保達, Yuanda 願達, Fashen 法深, Song *Falü* 宋法律, Baohui 保會, Baosheng 保勝, Jingsheng 淨勝, Yuansheng 願勝, Guangjin 廣進, 14 monks total. Of those, at least 5 monks are "Jingjie, Yuanzhen, Baoda, Yuanda, and Fashen" in P. 2049V (2) "Expenditure Record of the Head of Farming of the Jingtu Monastery for the Second Year of Changxing in the Later Tang (931)," so it can be confirmed that this monastery is the Jingtu Monastery, and the year of Bingshen here is the year 936, so this document titled "Record on Distributing Scriptures Regarding Lecturing on the 'Mahāpranjāpāramitā Sutra' by the Jingtu Monastery in the Bingshen Year (936)".

The size of the Mahāpranjāpāramitā Sutra is 600volumes, and since 14 monks read 310 volumes, so although this document is not complete, we know that there should only be 14 monks responsible for chanting on this occasion. According to P. 2049V (2) which recorded the signatures of the disciples of the Jingtu Monastery, we know that in 931, the number of monks in the Jingtu Monastery was 25. The size of the monastic establishment in 936 did not change significantly from that. Therefore, the proportion of those participating in the scripture chanting is about 56% (14/25).

In comparison, when analyzing the notes in the signature list of P. 2049V (2) above, at least seven monks with crude writing, such as "Bao", "Dao", "Ying", and "Yinhui", did not appear in the scripture chanting list on this occasion. This shows that their literacy level is indeed limited, and they are not competent enough for the religious activities that require a long time to read the scriptures. The monks who had a high level of hand writing in the Jingtu Monastery in 931 and the monks who were able to read the scriptures in 936 should be of the same group. This is no coincidence, and it also shows that they should be all the literate people in the Jingtu Monastery at that time.

(6) P. 3365 "Record on Distributing Scriptures to the Monks of the Jingtu Monastery for the Minor Illness of the Great King, Lord of the Prefecture on the Tenth Day of the Fifth Month of the Jiaxu Year (974)"(*jiaxu nian wuyue shiri jingtusi sengzhong wei fuzhu dawang xiaohuan fujing li* 甲戌年 (974) 五月十日淨土寺僧衆為府主大王小患付經歷).

The beginning and end of this document are both complete. It records the distribution of scriptures to the monks of a certain monastery praying for the Great King for his well being. The monks involved in the scripture chanting include Monk Zhou 周和尚, Monk Li 李和尚, Suo *Falü* 索法律, Wang *Falü* 王法律, Li *Falü* 李法律, Jie *Falü* 捷法律, Master Tan 譚法師, Master Gao 高法師, Gao *Falü* 高法律, Tan Sheli 譚闍梨, Fuman 福滿, Luo Laosu 羅老宿, Sengnu 僧奴, Jieyong 戒顒, Jieguang 戒光, Jiesong 戒松, Baofu 保福, Yingqi 應啓, Zhangsan 章三, Fajin 法進, Zhifa 智法, monastery governor Zhou (*Zhou sengzheng* 周僧正), 21 monks total.

In 974, Cao Yuanzhong 曹元忠, the Military Commissioner of the Guiyi Army, died of illness. Before his death, the Dunhuang monasteries held a scripture chanting activity to pray for him. This document is a record of the distribution of scriptures for the monks of a certain monastery who chanted the scriptures for Cao Yuanzhong in 974. Among them, "Monk Zhou (monastery governor Zhou), Tan *Falü*, Li *Falü*, Suo *Falü*, Jie *Falü* (Choujie), Zhangsan" are also found in S. 6452 "Expenditure Record of the Jingtu Monastery from the Xinsi Year to the Renwu Year (981–982)". Therefore, the monks in this document should be the disciples of the Jingtu Monastery. S. 6452 records the economic activities of the Jingtu Monastery from the twelfth month of the Xinsi year to the twelfth month of the Renwu year. Based on this, we can find the people who have economic relations with the monastery. In the expenditure record, if monks in other monasteries were recorded, the simplified name of the monastery to which the monk is affiliated will be added before the monk's name. For example, on the sixth day of the third month of the Renwu year, some millet "were loaned by Baotong 保通 of the Dacheng Monastery," so those monks who did not add the name of their monastery should have been the monks of the Jingtu Monastery. According to my estimates, there are at least 33 monks in the Jingtu Monastery who were involved.

There are a total of 21 monks in this case, so if the 33 people are used as the benchmark, the ratio of those who read the scripture is 63.6% (21/33).

We obtained 16 pieces of data based on the 13 source materials above. Here, taking the monastery as the unit, arranged by order of time, is as follows (see Table 7):

**Table 7.** The literacy rate of monasteries and convents in Dunhuang.

| Monastery | Time | Literacy Rate | Source | Type of Source |
|---|---|---|---|---|
| Lingtu Monastery | 788 | 76.5% | S. 10967 | name list for scripture chanting |
| | 893 | 66.7% | P. 3100 | Signature list |
| | 905 | 75% | S. 2575 (3) | Signature list |
| Jinguangming Monastery | Around 810s–820s | 84.6% | S. 2711 | name list of monks copying scriptures |
| | 841 | 88.5% | P. 3730 | Signature list |
| Longxing Monastery | 831 or 843 | 82% | P. 3947 | name list for scripture chanting |
| Dayun Monastery | 985 | 63.6% | 羽52 | Signature list |
| Jingtu Monastery | 925 | 61.9% | P. 2049V (1) | Signature list |
| | 931 | 64% | P. 2049V (2) | Signature list |
| | 936 | 56% | P. 2680V (8) | name list for scripture chanting |
| | 974 | 63.6% | P. 3365 | name list for scripture chanting |
| Lingxiu Convent | 788 | 34.3% | P. 3060 | name list for scripture chanting |
| Puguang Convent | 788 | 31.9% | P. 3060 | name list for scripture chanting |
| Dacheng Convent | 788 | 43.2% | P. 3060 | name list for scripture chanting |
| Total for Nuns | 788 | 32.7% | P. 3060 | name list for scripture chanting |
| Anguo Convent | End of 9th Century to the Beginning of the Tenth Century | 28.8% | S. 11352 | name list for scripture chanting |

Here, we also need to discuss two issues. The first is the reliability of the data, and the second is how representative are the data.

First, let us look at the reliability of the data. Judging from the above statistical table, although the literacy rate data of a specific monastery within a short period were obtained from different source materials, the data were still very similar. For example, in the relevant data of Jinguangming Monastery, S. 2711 and P. 3730 were the name lists of copying the scriptures and the signatures respectively. The figures we obtained are relatively close, being 84.6% and 88.5% respectively; in the ten years from 925 to 936, in the three sets of data from Jingtu Monastery (61.9% in 925, 64% in 931, and 56% in 936), two of the sets were gotten from the signature and one set of data from the name list for scripture chanting, which belong to different types of source materials, but the data are also relatively similar. The similarity of the data obtained from different source materials shows that the above statistics are relatively reliable.

It should be pointed out that signature is only a kind of "functional" literacy, while copying and reading scriptures is "full literacy". There are more monks who have the ability to sign than those who can copy and read scriptures. This explains why the data obtained from different materials in the same period are relatively close, but the data obtained from the signature list is higher than the data obtained from the name list of monks copying scriptures and the name list for scripture chanting. For example, regarding the three sets of data from the Jingtu Monastery, the data for the years 925 (61.9%) and 931 (64%) obtained from the signature lists were higher than the data for 936 (56%) obtained from the name list for scripture chanting.

Second, we will take a look at how representative are the data. Among the 16 pieces of data obtained, only 1 is the data pertaining to all the nuns, and the others are all data of individual monasteries; and the distribution of data for different monasteries

is also unbalanced. During the Tibetan and Guiyi Army periods, there were as many as 18 monasteries and convents in Dunhuang. However, this paper only used the data for 6 monasteries, especially the Jingtu Monastery (4 data) and the Lingtu Monastery (3 data). The discussion in this article is based on a specific monastery, so, can the data of a single monastery represent the literacy level of the entire Dunhuang sangha during the same period?

Among the data obtained from different materials, there are four sets of data involving Lingtu Monastery (76.5%), Jinguangming Monastery (84.6% and 88.5%), and Longxing Monastery (82%) in the middle and early stages of the Tibetan rule. They are very close in number and were all around 80%. This shows that the literacy level of monks in all the monasteries at that time was very high, and it also shows that 80% may also be the basic level of most of the monasteries with monks. During the Guiyi Army period, the seven data figures of different monasteries were all around 60%, which also showed that the literacy levels of these monasteries were very close to each other.[12] Therefore, if the overall literacy rate of male monks in Dunhuang during the same period is estimated based on the literacy rate of one monastery, even though there might be some deviation, the deviation may not be very large.

On the basis of the above analysis, the literacy rates of male monks in Dunhuang in the late Tang, Five Dynasties and early Song periods are arranged as follows (see Table 8):

**Table 8.** The literacy rate of Dunhuang monks in the period of Tibetan occupation and the period of the Guiyi Army.

|  |  | Time | Literacy Rate |
|---|---|---|---|
| Period of Tibetan Occupation | 1 | 788 | 76.5% |
|  | 2 | Around 810s–820s | 84.6% |
|  | 3 | 831 or 843 | 82% |
|  | 4 | 841 | 88.5% |
| Period of the Guiyi Army | 5 | 893 | 66.7% |
|  | 6 | 905 | 75% |
|  | 7 | 925 | 61.9% |
|  | 8 | 931 | 60% |
|  | 9 | 936 | 56% |
|  | 10 | 974 | 63.6% |
|  | 11 | 985 | 63.6% |

Regarding nuns, we have also only collected 6 pieces of data, of which there are fairly numerous data from 788, including the data on the Lingxiu Monastery, Puguang Monastery and Dasheng Monastery. If the overall figures of the three monasteries are taken, they can be arranged in chronological order as follows (see Table 9):

**Table 9.** The literacy rate of Dunhuang nuns in the period of Tibetan occupation and the period of the Guiyi Army.

|  |  | Time | Literacy Rate |
|---|---|---|---|
| Period of Tibetan Occupation | 1 | 788年 | 32.7% |
| Period of the Guiyi Army | 2 | End of 9th Century to the Beginning of the Tenth Century | 28.8% |

And We can put the above data in a chart (see Chart 1):

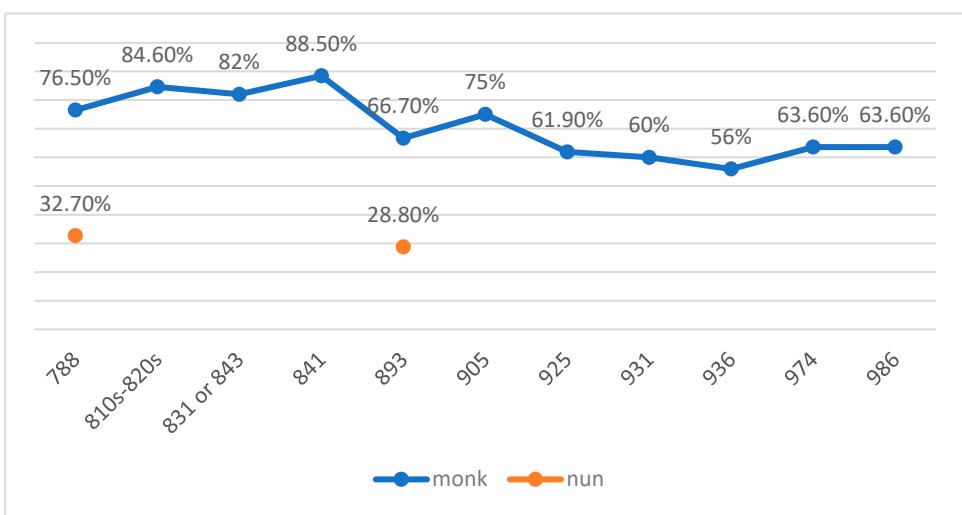

**Chart 1.** The literacy rate of Buddhist monks and nuns in Dunhuang during the late Tang, Five Dynasties, and early Song Period.

Although there may be deviations between the estimations of specific years and the real situation, and the specifics may be adjusted with further research, while the general trend is very clear: that is, the literacy rate of the sangha during the Tibetan period was significantly higher than that of the Guiyi Army period, and the literacy rate of the monks was higher than that of the nuns. At the same time, no matter how the state policy changed, the sangha could guarantee a certain number of literate monks, and this ratio was fairly high.

## 4. Reasons That the Literacy Rate of the Sangha in Dunhuang Declined under the Guiyi Army Period

During the Tibetan occupation period and the Guiyi Army period, the environment for the development of Buddhism was very different. During the Tibetan occupation period, Buddhism in Dunhuang was isolated from the Central Plains and was relatively closed off, gradually showing a different development from Buddhism in the Central Plains. During the Guiyi Army period, with the return of Dunhuang to the Tang Dynasty, communication between Dunhuang and Central China became intensified, and Dunhuang Buddhism gradually converged more and more with Buddhism in the Central Plains. The changes in the literacy rate of the sangha should be considered under this historical context. Here, we focus on three factors. The first is the change in the policy of ordaining monks and the expansion of the size of the sangha. The second is the change in the number of eminent monks, and the third is the change in the Buddhist ethos.

(1) The Change in the Policy of Ordaining Monks and the Expansion of the Size of the Sangha.

To a great extent, the state's Buddhist policy, especially the policy of ordaining monks, determines the education level of the Sangha. For example, in the early Tang period, the policy of "test on the sutras to ordain the monks" (*shijing* 試經) was widely implemented.[13] That is, monks were required to read hundreds of scriptures to obtain the qualification of ordination. Furthermore, from the time of Emperor Gaozong, daily examinations were often practiced (Meng 2009, pp. 136–43). When the policy is effectively implemented, the official can guarantee the quality of the Sangha. If monks were ordained indiscriminately, especially when monk identifications were sold excessively, the education level of the Sangha will drop sharply.

Although it is not clear what the local policy pertaining to ordaining monks in Dunhuang was during the Tibetan occupation period and the Guiyi Army period, it can still be inferred from the analysis of the changes in the size of the Dunhuang sangha that the government of the Tibetan period strictly controlled the monks, while the Guiyi Army

gradually loosened the control. Before the third month of 788, the size of the sangha was 310. Later, due to the immigration of monks outside of Dunhuang, the size may have reached 389; by 825, it was 427 (218 male, 209 female). Although the size was also increasing at the time, it was not a very large increase. During the period of the Guiyi Army, the size of the sangha expanded rapidly. The size was 1169 at the beginning of the 10th century. Among them, there were 392 male monks, which almost doubled in the size compared to the year 825. The number of nuns reached 777, an increase of more than three times. In 933, the size of the sangha had reached 1500, 3.5 times that of 825. During the period of the Guiyi Army, it was obvious that the quality of the sangha could not be effectively controlled through the policy of the ordination of the monks.

The relaxation of restrictions on the size of Buddhist sangha meant that the quality of the sangha at the institutional level cannot be guaranteed. At the same time that this caused the expansion of the size of the sangha, education within Buddhist institutions would have taken a huge hit. During the period of the Guiyi Army, the expansion of the size of the sangha far exceeded the speed of growth in the other various fields of the monasteries' construction. This increasingly limited the resources of the monastery for matters such as Buddhist practices and monks' livelihood.

For example, in terms of monastic education, the educational resources of monasteries may not have achieved the same growth as the size of the sangha. Although some monks in Dunhuang received a good family education before they became ordained, for the vast majority of monks, their education training in aspects such as reading and writing was completed in the monastery. During the Tibetan and Guiyi Army periods, there were some male monks' monasteries, such as temple schools (*sixue* 寺學), which could be open to some male monks, and the new monks, śrāmaṇeri, and śikṣamāṇās, could learn from older monks and nuns, but many source materials show that during the Guiyi Army period, the cultural training received by many monks was very inadequate. The document P. 6005, "Announcement About the Summer's Three-month Retreat During the Guiyi Army Period" (*guiyi jun shiqi shimen tie zhusi gangguan ling xiaanju tie* 歸義軍時期釋門帖諸寺綱管令夏安居帖) mentioned that many monks, śrāmaṇeri, and śikṣamāṇās "did not have teachers yet" (weiyou qing yizhi 未有請依止) at that time, that is, there was no elder monks to guide them in their cultivation. In such an environment, the condition of their training in cultural matters can be imagined. In the signature lists of the later periods of the Zhang family's Guiyi Army and the Cao family's Guiyi Army period analyzed above, there are a lot of crude signatures, and this phenomenon is not seen in the signature lists of the Tibetan period. In another example, for the year 925, six monks in the Jingtu Monastery, including "Bao", "Dao", "Ying", and "Yinhui" have their own signatures in the document P. 2049V (1). At this time, their hand writing was very crude, and they were obviously beginners. Judging from their signatures on P. 2049V (2) in 931, in six years, only the hand writing of "Bao" has improved slightly, and the writing ability of the five other people did not improve compared to before. Some of their writing abilities even regressed, which means that these five monks had little to no writing training in the past six years. (See Table 10).

**Table 10.** A comparative table of calligraphy by Bao and five others in 925 and 931.

| | **Bao**寶 | **Dao**道 | **Ying**應 | **Baoda**保達 | **Yinhui**因會 | **Gu**古**(?)** |
|---|---|---|---|---|---|---|
| 925 |  |  |  |  |  |  |
| 931 |  |  |  |  |  |  |

The rapid expansion of the size of the sangha has also far exceeded the construction speed of the basic infrastructure of the monastery such as dormitories. As a result, a

large number of monks cannot live in the monasteries and can only return to their secular families (Hao 1991, pp. 836–37; 1998, pp. 74–112; Wu 2018, pp. 14–21). Staying at secular home meant that the studies in the Buddhist subjects of sutras, vinaya, and Buddhist treatises were in a very bad situation. S. 371 and P. 3092V are "An Announcement about the Examination from the Monastery Education Establishment in the Tenth Month of the Wuzi Year (928)" (*wuzi nian* [928] *shiyue shibu tie* 戊子年 (928) 十月試部帖), which recorded that in 928, the monastery education establishment ordered the monastery managers to supervise their disciples twice a month (the first and last day of the month) in reading and reading sutras, vinayas and Buddhist treatises. In order to ensure these requirements are carried out, regulatory measures such as setting up "teaching masters" to teach, having "Karmadāna making reports" and having "examination on scriptures" were implemented. Even so, the chanting that resulted from the remaining 20 people found in P. 3092V show that 8 people participated in all the activities, 3 people participated in half of the activities, and 9 people did not participate. Nearly half of the disciples ignored the announcement. The appearance of this phenomenon indicates that the monks at that time might not take this kind of scripture examination very seriously. This attitude will inevitably lead to a sharp drop in the monks' education level compared to the past. At the same time, it is worth noting that the proportion of monks who participated in chanting in 928 was 55% (11/20), which is almost the same as the above statistics of 56% (P. 2680V [8]) of the proportion of those chanting in the Jingtu Monastery in 936.

(2) Decrease in the Number of Eminent Monks and the Transformation of the Buddhist Ethos

The literacy rate of the monks in Dunhuang during Tibetan rule was much higher than that during the Guiyi Army period, which was also closely related to the level and atmosphere of Buddhist studies in Dunhuang.

The rulers of Tibetans placed great emphasis on Buddhism, and once adopted the policy of besieging and not attacking the city of Dunhuang for more than ten years. This allowed Buddhism in Dunhuang to avoid a military disaster to the greatest extent. At the same time, Dunhuang also obtained a large number of Buddhist scriptures and Buddhist monks from places such as Ganzhou. It was also during this period that famous monks such as Tankuang 曇曠, an eminent Vijñānavāda monk from Ximing Monastery in Chang'an, retreated to Dunhuang to spread the Buddhist teaching. As Rong (2015) has already pointed out, "during the period of Tibetan rule (786–848), Buddhism in Dunhuang developed rapidly. The number of monasteries, monks and nuns continued to increase. Organized scripture copying led to the enrichment of scriptures stored in monasteries. Eminent Han and Tibetan monks such as Tankuang, Mahayana Hoa-San 摩訶衍, and Facheng 法成, either concentrated on writing, on spreading meditation methods, or on translating scriptures and lecturing, which lifted the level of Buddhist teaching in Dunhuang to an unprecedented level". (Rong 2015, p. 268). In the early period of the Zhang family's Guiyi Army, under great monks such as Facheng, Hongbian, Wuzhen, and Fajing 法鏡, Buddhist studies in Dunhuang were maintained at a relatively high level. Examples include Cheng'en's *Commentary on the Mahāyāna-śatadharma-prakāśamukha-śāstra*, (*baifa lunshu* 百法論疏) which was even approved by the great monks of Chang'an. However, during the period of the Cao family's Guiyi Army, there was never another eminent monk who could compare with Tankuang and Facheng, and there was never another work that could be compared with the *Commentary on the Yogācārabhūmi-śāstra* (*yujia shoui* 瑜伽手記) and *Commentary on the Mahāyāna-śatadharma-prakāśamukha-śāstra*, and even the scene where "one monk taught the Dharma, and all the monks gathered together (to listen)" was rarely seen.

There may also be some connections between the decline of the education level of monks and the strength of the influence of different Buddhist schools in the Dunhuang region. Monks like Tankuang and Facheng took Vijñānavāda learning as their doctrine, and with their passing away, Vijñānavāda learning gradually declined. At the same time, during the Guiyi Army period, with the strengthening of the connection between Dunhuang and the Central Plains, the influence of Buddhism from the Central Plains became increasingly

prominent. The more simple and easy to practice Pure Land Buddhism from the Central Plains, featuring chanting rhythmic Buddhist songs and the name of Buddha, is increasingly becoming the mainstream teaching in Dunhuang.

Compared with the Vijñānavāda school's emphasis on knowledge of Buddhist theories, the Pure Land school pays more attention to the mastery of techniques such as lecturing and singing. Such as many hymns collected in S. 2945 "*Rituals of Pure Land on Chanting Buddha, Chanting Scriptures through Five Pronunciation Techniques*" (*jingtu wuhui nianfo songjing guanxing yi* 淨土五會念佛誦經觀行儀) are marked with harmony terms such as "How happy the pure land is (*jingtu le* 淨土樂)". From this, it can be seen that in the ritual of Pure land Buddhist, the great master leads the singing, and the disciples sing harmonious terms along in unison. In this way, ordinary monks only need to memorize some simple words for chanting. Furthermore, the Pure Land rituals used mostly Buddhist songs. They have rhythms and rules, which are catchy when chanted, and their theories are easy to understand and convenient to memorize. Therefore, it is easier for the monks to operate this, and they only need to listen and sing constantly without the need to read the texts to master it. In reality, for some performing monks, though they may not be literate, they can still master enough performance songs to meet the needs of the Buddhist rituals. This reduces their demand for profound knowledge of Buddhism, which also lowers their level of education.

## 5. The Significance of the Decrease in the Literacy Rate of the Sangha during the Period of the Guiyi Army from the Perspective of the Monks Staying at Home

During the Guiyi Army period, the decrease in the literacy rate of the Dunhuang sangha obviously had huge negative consequences. For example, it would cause the sangha's status in the relationship between the government and religion to further decline,[14] and the gap between the rich and the poor within the sangha would widen.[15] However, under the background of the rapid expansion and the secularization of Buddhism, the number of literate monks was also increasing. They were more closely integrated with the secular society, and the literate monks were obviously of positive significance to the regional society.

When the policy of "test on the sutras to ordain the monks" failed, and it was impossible to ensure the education level of the monks from the administrative system of the state, the maintenance of the literacy rate of the monks at a certain level was largely the result of the self-sustaining effort of the monks themselves. The sangha is a religious group as well as a cultural group. Internally, there was an educational system that crossed over with ancient secular education, but is quite independent of it. Even if the state could no longer guarantee the quality of the sangha from the outside, this system can still ensure that some illiterate groups can grow into qualified monks after entering the monastery. The cultural resources of the monasteries can even meet the needs of many secular scholars. Even up to the middle and late Tang period, there were still many secular scholars who went to study in monasteries (Yan 1969, pp. 367–424). Dunhuang was even more obvious in that the monastic schools opened by monasteries even became the most important education center in the prefecture at one point (Li 1986, pp. 39–47; Gao 1986, pp. 231–70). Therefore, even when the sangha group was viciously expanded and the overall education level of the ordained monks was very low, the monastery could still support the development of a considerable number of disciples. Specifically in Dunhuang, the resources of monasteries with monks can also maintain the development of a good half or more of the monks, and it is reflected in the fact that the literacy rate of the monks was still about 50–60%. Although the resources of convents were more limited, it could also ensure that about one-third of the disciples obtained a certain degree of knowledge.

Therefore, although the size of the Dunhuang sangha during the Guiyi Army period was expanding rapidly, the number of literate people was also increasing; furthermore, in the context of the decline in the total population coupled with the expansion of the sangha at this time, the proportion of literate monks in the total population is actually also

rising. In the early days of Tibetan rule, Dunhuang had a total population of more than 30,000 people (Qi 1989, pp. 73–97), although there were only 180 literate monks (including 103 monks and 77 nuns), accounting for 0.6% of the total population. By the Guiyi Army period, this proportion continued to rise. At the end of the ninth century and the beginning of the tenth century, the literacy rate of monks mostly remained at about 50–60%, and even the literacy rate of nuns, whose growth had been out of control by the beginning of the tenth century, was about 28.8%. If the literacy rate of the monks in 893 was 66.7%, and the literacy rate of the nuns in the early tenth century was 28.8%, based on the data on the size of the Buddhist establishment of Dunhuang in the early tenth century (392 monks and 693 nuns) obtained from S. 2614V, the number of literate people in the Buddhist establishment may be 261 (392 × 66.7%) + 224 (777 × 28.8%), which is 486. Compared with the early days of Tibetan rule, the number of literate monks increased by 158, and the number of nuns also increased by 142; both of them more than doubled. At the end of the ninth century, the population of Dunhuang under the rule of Zhang Chengfeng was more than 10,000, and the literate people in the sangha accounted for nearly 5% of the total population! Considering that before modern times, the literacy rate of the total population was only about 10% (See Jack 1963, pp. 304–5; John 1983, pp. 572–99), this figure is very high. Of course, the reason why this value looks so surprising is due more to the frequent wars during the Zhang Chengfeng period, which resulted in a large loss of population. However, even using the population of 30,000 during the Cao family Guiyi Army period for estimation, the proportion of literate monks can still reach about 1.5%, which is much higher than that during the Tibetan period. Erik Zürcher once described the ancient Buddhist sangha as "the Secondary Elite" (See Erik 1989, pp. 19–56). From the scale of the literacy rate, his description is very accurate. The increase in the number of literate monks and their increase in proportion in the total population means that the size and proportion of the population they can influence with their knowledge has also increased.

Although the literate monks of the sangha during this period were likely to increase in quantity rather than in the level of their literacy, it can not be denied that these literate monks played the dual role of holding religious authority and cultural authority at the same time. In local societies, they are among the most important users of knowledge. In fact, judging from the contents that were studied daily by the Dunhuang monks, in addition to Buddhist knowledge, they also actively learned and mastered knowledge related to secular affairs. From the writing practice texts of the monks in Dunhuang, scholars like Pei Changchun 裴長春 and Shen Shoucheng 沈壽程 found that those documents of social organizations (*sheyi wenshu* 社邑文書), contract documents, and letters accounted for a large proportion of the monks' daily learning (Pei and Shen 沈壽程 2020, pp. 29–37).

The lifestyle of many monks is to live in secular families, which will allow what they learn in the monastery to influence many secular people who live outside the monastery. So the monks would then play an important role in maintaining the normal operation of the regional society.

For example, the *sheyi* 社邑 was a kind of community organization where people came together voluntarily to help each other in religious and social activities, which played an important role in maintaining the normal operation of rural society. Many of them had the participation of monks. There are 18 *sheyi* articles (*shetiao* 社條) included in the "Compilation and Commentary on the Documents of Dunhuang *Sheyi*", (*dunhuang sheyi wenshu jixiao* 敦煌社邑文書輯校). Their dates are concentrated after 855, and 10 of them contain the members' name. Furthermore, 7 out of the 10 documents contain the monks or nuns' names. This also means that about 70% of the community organizations have monks or nuns in them. Moreover, many monks often play important social roles, not just as ordinary members, but holding the most important position as the "three officials" in a *sheyi* (head of *she* 社長, official of *she* 舍官, and the recorder 錄事). P. 4960 "Articles on Building the Buddhist Hall *she* Concluded after Selection of the Three Officials on the Twenty First Day of the Fifth Month of the Jiacheng Year [944]" (*jiacheng nian* [944] *wu yue ershiyi ri xiu fotang she zaiqing sanguan yue* 甲辰年 (944年) 五月廿一日修佛堂社再請三官約)

are new articles concluded by the Buddhist Hall Association (*fotang she*) after the re-election of the three officials. Among them, the three elected officials were all monks: "Qingdu 慶度 is the official of *she*, Fasheng 法勝 is the head of the *she*, and Qingjie 慶戒 is the recorder". In S. 6005 "Supplementary Treaty of a *She* in Dunhuang" (*Dunhuang moushe buchong tiaoyue* 敦煌某社補充條約), which was written in the first half of the tenth century, the two elders of the *she* (*shelao* 社老), "Xici 喜慈, Wenzhi 文智", and others were all monks. Being an official of *she* shows that the monks had prestige in society. On the other hand, it also shows that these monks had sufficient ability, which of course also included knowledge. This is especially true in the position of "recorder", which was mostly "held by those who are capable, smart and upright. They take care of the daily tasks of the organization, such as posting articles, organizing Buddhist gatherings, managing funerals, and supervising members to abide by the regulations and enforce penalties".[16] As a result, the knowledge required of them is even more evident.

Monks also played an important role in economic activities that are closely related to the local community such as the establishment of contracts. In the *Compilation and Commentary on Dunhuang Contract Documents* (*dunhuang qiyue wenshu jixiao* 敦煌契約文書輯校), in the "buy and sale category", there are 17 documents which contain information on things such as signatures, among which 8 documents mention monk participants (not buyers and sellers), making up 47.1% of the total documents (See Sha 1998, pp. 1–81). The monks would act as middlemen (*zhongren* 中人) or witnesses (*jianren* 見人) and would undertake the task of writing contracts. In P. 3394 "Land Contract Between the Monk Zhang Yueguang and Lü Zhitong in the Sixth Year of Dazhong of the Tang (852)" (*tang dazhong* [852] *seng zhang yueguang, lüz zhitong yi diqi* 唐大中六年 [852] 僧張月光, 呂智通易地契), the first witness listed is "Monk Zhang Fayuan 僧張法源," followed by "Monk Shanhui僧善惠". Furthermore, Fayuan also signed the contract, indicating that he has a certain level of education. The "contract writing person" (*shuqi ren* 書契人) in S. 1475V "Contract Pertaining to Wheat With the Resident of the Stong sar Tribe, Zhai Milao in the Year of The Year of the Rabbit (823) (*maonian* [823] *xidongsa buluo baixing zhai milao bian mai qi* 卯年 [823] 悉董薩部落百姓翟米老便麥契), is "Monk Zhizhen僧志貞". The document BD 3925V (11) "Contract Pertaining to the Resident of the township of Mogao, Zheng Chouda Selling Houses in the Ninth Years of Kaibao (976)" (*kaibao jiunian* [976] *mogao xiang baixin zheng chouda mai zhaishe qi* 開寶九年 [976] 莫高鄉百姓鄭丑達賣宅舍契) was also written by Monk Zhijin 僧志進.

In addition, in the Dunhuang documents, there are also many monks' writings on documents on releasing wives (*fangqi shu* 放妻書), documents on releasing slaves (*fangliang shu* 放良書), wills, documents on brothers dividing property (*xiongdi fenjia shu* 兄弟分家書), texts on rituals for childbirth, and texts on divination, which seem to have covered all aspects involving writing in the daily lives of ordinary people. We can see that although the knowledge of many literate monks was obtained through the Buddhist education system, they can use this knowledge for the livelihood of the people in the local society, and their influence was so comprehensive and deep that they became the maintainers of the normal operations of the local society. When considering this, although the development of Buddhism in Dunhuang during the late Tang and Five Dynasties was relatively bleak, the role the sangha played in regional society, especially the secular society, may have been much greater than that in the early Tang period.

## 6. Conclusions

Through the signature list of monks, the name list of monks copying scriptures and the name list of monks chanting scriptures, this paper has made a relatively detailed statistics on the literacy rate of the Dunhuang monks in the late Tang, Five Dynasties and the early Song period. Although these data are based on local texts in Dunhuang, they can also serve as reference for the literacy of Buddhist sangha in the Central Plains. For example, in the early days of the Tibetan occupation of Dunhuang, especially when the S. 2729 (1) was created in the third month of 788, the Dunhuang sangha was actually completely inherited

from the Tang period. This also means that after the An-Shi Rebellion 安史之亂, the literacy rate of the monks in Dunhuang should have been around 76.5%. Before that, Dunhuang was still a standard prefecture of the Tang Dynasty. Zhang Yichao 張議潮 later overthrew Tibetan rule and brought Dunhuang back under the Tang Dynasty. Although the Guiyi Army regime was quite autonomous, the Buddhist policies, trends of Buddhism, and the living situation of the sangha at that time were actually similar to those of the Central Plains. Therefore, the literacy rate of the Dunhuang sangha at this time should also serve as an important reference for understanding the development of Buddhist in the Central Plains.

On this basis, we can put forward a new understanding of the role the sangha played in the development of regional society in the late Tang, Five Dynasties and the early Song period and the development trend of Buddhism after the Song Dynasty.

The rapid expansion of the sangha in the Tang Dynasty appeared after Emperor Daizong 代宗, and reached a size of 260,000 monks and nuns before the Huichang Persecution of Buddhism (*huichang fanan* 會昌法難). This is over twice the size of the 126,000 monks and nuns in the twenty-fourth year of Kaiyuan (736) under Emperor Xuanzong 玄宗.[17] In the period of Xuanzong's reign, even if all monks and nuns were literate, they only accounted for 0.25% of the total population; and even if the 260,000 monks and nuns under Wuzong's reign only had a literacy rate of 50%, the number of literate monks and nuns still reached 130,000, and their proportion in the total population increased to 0.5%.[18]

Similar to Dunhuang, throughout the Tang period, the living situation of monks in the Central Plains also experienced great changes. In the early Tang period, the government had strong control over the development of the sangha. The literacy rate of the sangha might have been relatively high, but its scale was limited, and the monks mainly lived in monasteries. Even though there appeared the phenomenon of monks who "lived in the secular disciples' family", (侍養私門)[19] it was quickly rectified. Overall, the degree of integration between the sangha and the life of the secular people was not very frequent and deep. Before Wuzong 武宗 persecuted Buddhism, there were also many monks who lived in secular homes and were neighbors with ordinary secular people in many places in the Central Plains. In a petition during Tang Dezong's 唐德宗 time, it was mentioned that many monasteries were occupied by military personnel at the time. The monasteries also stopped providing food to the monks (所在伽藍, 例無飯僧). Furthermore, a lot of monasteries even did not have a canteen (the content of the petition can be seen in P.3608V and P.3620). In monasteries without a canteen, monks obviously cannot stay there long term, and had to make a living themselves to survive. This is consistent with the characteristics of Dunhuang monasteries where they only "provide food when they have an event", (有事供粮) (Hao 1998, pp. 123–63). During the reign of Emperor Wenzong 文宗, Ennin 圓仁 also saw the phenomenon of "monks all living in secular homes" (僧盡在俗家) in the Beihai County of Shandong and other places.[20] When Wuzong persecuted Buddhism, almost all monks and nuns were forcibly returned to laity, allowing literate monks to return to the secular society. There were many highly knowledgeable people among them. The *Biography of Eminent Monks in the Song* 宋高僧傳 records that many eminent monks "wrapped their heads to become commoners" (裹首為民) when Buddhism was persecuted, and lived in secular society for several years, and many monks never returned to the monasteries even after Buddhism was restored in the period of the reign of Emperor Xuanzong 宣宗.

Compared with the previous time when they lived in monasteries and devoted themselves to Buddhist affairs, after walking out of the monastic gates and returning to the secular society again, the monks would also apply the knowledge they learned in the monasteries to their everyday secular life when they were at secular home. This is the background under which there was an increase in the secularization of Buddhism during the Song and Ming Dynasty, in which there was more and more involvement of Buddhism in the daily lives of the secular masses (Zhang and Ren 2015, pp. 119–30; Chen 2019, pp. 157–63).

**Funding:** The research is funded by Archaeology and History study fund, School of History and Culture, Shandong University.

**Data Availability Statement:** All the data are calculated in this article, and there is no link.

**Conflicts of Interest:** The author declares no conflict of interest.

## Notes

1    T 396: 1119a.

2    T 2103, 24: 279a-b.

3    For a study, see Cheng (2019, p. 40).

4    Regarding the size of the sangha, please refer to Akira (1959, pp. 285–338). Chuguyevsky (2000, pp. 116–39). Zheng (2004a, pp. 20–30).

5    For functional literacy and full literacy, see Evelyn (1979).

6    In the entire list, only 8 monks, "Jinshu 金樞, Lisu 利俗, Liming 離名, Jieran 戒然, Fayuan 法緣, Qikong 晉 空, Chao'an 超岸, Tanhui 曇惠" are found in S. 2729 (1) and P. 3060. It can be seen that the time here might have been about twenty or thirty years from the records in S. 2729 (1). Therefore, overall, the age of this document is roughly from the 810s to the 820s.

7    For the relationship between these two written scrolls, see Zhao (2013).

8    For related controversies, see Liu (2017).

9    *Ryō no shūge* 8. 232–33. When Zheng (2004b) recovered the article "holding the position of the three directors" in *Regulations of Buddhist Monks and Daoists*' (*seng dao ge* 僧道格), he did not recover the text "their names needs to be signed by themselves in a report and send to the officials". However, according to the Dunhuang documents, the disciples did jointly sign it.

10    In addition to the 6 pieces analyzed in this article, P. 5587 (4) "The Report from Kaiyuan Monastery on the Fourth Month of the Year of Ox (809 or 821)" (*chounian* [809 *huo* 821] *siyue kaiyuan si die ji du sentong pan* 丑年[809或821] 四月開元寺牒) and the 羽 64 "Contract of Li Shanshan Selling His Houses to the Dayun Monastery in the Early Tenth Century" (*shi shiji chu li shanshan mai she yu dayun si qi* 十世紀初李山山賣舍於大雲寺寺契), S. 6417 (20) "The Report Pertaining to the Elder Shenwei and Others of the Jinguangming Monastery Inviting the Shanli to Fill the Position of Elder in the Third Month of the Second Year of Qingtai in the Later Tang Dynasty (938)" (*houtang qingtai er nian* [935] *sanyue jinguangming si shangzo shenwei deng qin shanli wei shangzui zhuang* 後唐清泰二年 (935) 三月金光明寺上座神威等請善力為上座狀)," S. 1625 "Report on the Estimation of the Expenditure Record of the Dasheng Monastery in the Sixth Day of the Twelve Month of the Third Year of Tianfu" (*tianfu sannian* [938] *shi'er yue liu ri dasheng si suan hui die* 天福三年 (938) 十二月六日大乘寺入破曆算會牒) and BD 14670 "Report of the Disciples of the Lingtu Monastery Nominating the Head of Registry in the Second Year of Guangshun (952)" (*guangshun errnian* [952] *lingtu si tuzhong ju gangshou die* 廣順二年 (952) 靈圖寺徒眾舉綱首牒), also had signatures from the monks, but the number of signatures are few or there were too many incomplete ones, and they have no reference value, so this article will not discuss them for the time being.

11    P. 3060 "Record of Scripture chanting by the Sangha on the Third Month of the Year 788" (788 *nian sanyue dunhuang sengtuan zhuanjing li* 788年三月敦煌僧團轉經曆), BD 16453 "Record of the Scripture chanting Rearding by the disciples of Lingxiu Monastery in the Early 11th Century" (*shiyi shiji chu lingxiu si zhuanjing li* 十一世紀初靈修寺轉經曆) also recorded the number of people who chanted scriptures in the monastery, but compared with the size of the monks in the monastery at that time, the number of participants was very small, and they do not have much reference value. Therefore, they will not be discussed here.

12    In the Tibetan and the Guiyi Army periods, the implementation of various policies might have maintained a relative equilibrium in the development of the monasteries in Dunhuang, such as the distribution policy regarding newly ordained novice monks. These newly ordained novice monks will all be given the dharma name and the same group of monks often had the same generation name. This can be seen from P. 3423 "Record of the Mitzvah for the Newly Ascended Monk in the Qianyuan Monastery," (*bingxu nian* [926] *qianyuan si xindeng jieseng cidi li* 丙戌年 (926年) 乾元寺新登戒僧次第曆). However, they could not choose a monastery based on their own preference, but were most likely uniformly distributed to different monasteries by the government. Therefore, in S. 2729 (1), monks such as "Jinluan 金鸞, Jinyun 金雲, Jingu 金鼓, Jinzhen 金振, Jinye 金液, Jindong 金洞," were probably also ordained in the same year, but were distributed to different monasteries later. The distribution mechanism is not clear, but this mechanism should not lead to much disparity between different monasteries. In addition to this, the flow of monks between different monasteries also guaranteed to a certain degree of balance in the development of the different monasteries. An example is found in the document P. 4660 (45) "Praise-Text of the Atcharya Xuan,"(*shazhou shimen dujiaoshou Xuan sheli zanbingxu* 沙州釋門都教授炫闍梨讚並序) where Zhang Jinxuan, who already had "many disciples," when young, resided in the Jinguangming Monastery. Later he was invited by the Qianyuan Monastery and played an important role in the development of that monastery.

13    See discussions in Zhou (2008, p. 15).

14    Rong Xinjiang once observed that since the time of Tibetan rule, due to the increase in the power of Buddhism, the highest monks officials had great power, often ruling Dunhuang society together with local rulers, until the time of Zhang Chengfeng 張承奉, when the Guiyi military regime had completely surpassed the power of the clergy. He also believes that the emergence of

this phenomenon is not unrelated to the termination of the Buddhist teaching activities in Dunhuang, the reduction of the self education of monks and nuns, and the expansion of the size of the sangha. See: Rong (2015, p. 275).

[15]  Participating in ritual activities such as scripture chanting is an important means for monks to obtain an income. Hao (1998, pp. 332–66) has found that the largest number of monks who participated in these activities were often those from the upper echelons of the monastic community. Although this phenomenon may be related to the fact that the older monks have some power to deprive other newer monks of opportunities, it is also likely to be the result of the inability of the sangha to provide enough literate monks. In 936, there were 25 monks at the Pure Land Monastery, but only 14 monks, including the supreme Buddhist chief monk, participated in scripture chanting. Only 14 monks were responsible for reading the 600-fascicle *Mahāprajāpāramitā* Sutra, which was a very heavy task. So why were the monks such as "Bao", "Dao", "Ying", and "Yinhui" not able to participate? An important reason is that they were illiterate. After all, many ritual activities, such as scripture chanting, require the chanting of texts in sutras, and those who are illiterate were excluded. Therefore, the supreme Buddhist chief monk and others had to take turns. Therefore, from this perspective, the reduction of the overall education level of the sangha will also cause the stratification of the rich and poor within the sangha to be more serious to a certain extent.

[16]  For related discussions, see Ji (1998, p. 426).

[17]  *Tang liudian* 4: 125.

[18]  During the time of Wuzong's 武宗 reign, the number of households was 4,955,151, and the population was about 20 million. See: *Cefu yuangui* 159: 5515.

[19]  *Cefu yuangui* 159: 1775.

[20]  Nittō-guhō-junrei-kōki no kenkyū 入唐求法巡禮行記の研究, pp. 228–349.

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

### Secondary Studies

Akira, Fujieda 藤枝晃. 1959. Tonkō no sōni seki 敦煌の僧尼籍. *Tōhō gakuhō* 東方學報 29: 285–338.

Chen, Lei 陳雷. 2019. Songdai fojiao shisu hua de xiangdu ji qi ishi 宋代佛教世俗化的向度及其啓示. *Ningxia Shehui Kexue* 寧夏社會科學 5: 157–63.

Cheng, Minsheng. 2019. Lun songdai seng dao de wenhua shuiping 论宋代僧道的文化水平. *Zhejiang Daxue Xuebao* 浙江大学学报 3: 28–47.

Chuguyevsky. 2000. *Dunhuang Hanwen Wenshu* 敦煌漢文文書. Translated by Wang Kexiao 王克孝. Edited by Wang Guoyong 王国勇. Shanghai: Shanghai Guji Chubanshe.

Erik, Zürcher. 1989. Buddhism and Education in Tang Times. In *Neo-Confucian Education: The Formative Stage*. Berkeley: University of California, pp. 19–56.

Evelyn, Rawski. 1979. *Education and Popular Literacy in Ch'ing China*. Ann Arbor: University of Michigan Press.

François, Furet, and Ozouf Jacques. 1982. *Reading and Writing: Literacy in France from Calvin to Jules Ferry*. New York: Cambridge University Press.

Gao, Mingshi 高明士. 1986. Tangdai dunhuang de jiaoyu 唐代敦煌的教育. *Chinese Studies* 漢學研究 4: 231–70.

Hao, Chunwen 郝春文. 1991. Tang houqi wudai songchu shazhou sengni de tedian 唐後期五代宋初沙州僧尼的特點. In *Dunhuang Tulufan Xue Yanjiu Lunwen Ji* 敦煌吐魯番學研究論文集. Shanghai: Hanyu Da Cidian Chubanshe, pp. 817–57.

Hao, Chunwen 郝春文. 1998. *Tang houqi wudai song chu dunhuang sengni de shehui shenhuo*. 唐後期五代宋初敦煌僧尼的社會生活. Beijing: Zhongguo Sheke Kexue Chubanshe.

Jack, Goody. 1963. The consequences of literacy. *Comparative Studies in Society and History* 5: 304–45.

Ji, Xianlin 季羨林. 1998. *Dunhuang xue dacidian*. 敦煌學大辭典. Shanghai: Shanghai Cishu Chubanshe.

John, Baines. 1983. Literacy and Ancient Egyptian Society. *Man* 18: 572–99.

Li, Zhengyu 李正宇. 1986. Tangsong shidai de dunhuang xuexiao 唐宋時代的敦煌學校. *Dunhuang Yanjiu* 敦煌研究 1: 39–47.

Liu, Yonghua. 2017. Qingdai minzhong shizi wenti de zai renshi 清代民眾識字問題的再認識. *Zhongguo Shehui Kexue Pingjia* 中國社會科學評價 2: 96–110.

Meng, Xianshi 孟憲實. 2009. Lun tangchao de fojiao guanli—Yi sengji de bianzao wei zhongxin 論唐朝的佛教管理—以僧籍的編造為中心. *Beijing Daxue Xuebao* 北京大學學報 3: 136–43.

Mote, Frederick. 1972. China's past in the study of China today: Some comments on the recent work of Richard Solomon. *Journal of Asian Studies* 32.1: 107–20.

Pei, Changchun 裴長春, and Shoucheng Shen 沈壽程. 2020. Gudai sengren de zhishi jiegou—Yi dunhuang sengren xizi wenshuP. 2129V seng shanhui xizi wenshu weili 古代僧人的知識結構—以敦煌僧人習字文書P. 2129V《僧善惠習字文書》為例. *Yindu Xuekan* 殷都學刊 3: 29–37.

Qi, Chenjun 齊陳駿. 1989. Dunhuang yange yu renkou敦煌沿革與人口. In *Hexi Shi Yanjiu*河西史研究. Lanzhou: Gansu Jiaoyu Chubanshe.

Rong, Xinjiang 榮新江. 2015. *Guiyi Jun Shi Yanjiu* 歸義軍史研究. Shanghai: Shanghai Guji Chubanshe.

Sha, Zhi 沙知. 1998. *Dunhuang Qiyue Wenshu Jixiao* 敦煌契約文書輯校. Nanjing: Jiangsu Guji Chubanshe.

Tang, Genghou 唐耕耦, and Hongji Lu 陸宏基. 1990. *Dunhuang Shehui Jingjiwenxian Zhenji Shilu* 敦煌社會經濟文獻真跡釋錄 *No.4*. Beijing: Quanguo Tushuguan Wenxian Suowei Fuzhi Zhongxin.

Wu, Shaowei 武紹衛. 2018. Cong shehui jingji jiaodu kan tang houqi wudai songchu dunhuang sizhong jujia yuanyin 從社會經濟角度看唐後期五代宋初敦煌寺眾居家原因. *Zhongguo Shehui Jingji Shi Yanjiu*中國社會經濟史研究 3: 14–21.

Yan, Gengwang 嚴耕望. 1969. Tangren xiye shanlin siyuan zhi fengshang 唐人習業山林寺院之風尚. In *Tangshi Yanjiu Luncong* 唐史研究論叢. Hong Kong: Xinya Yanjiusuo, pp. 367–424.

Zhang, Zhuping 張祝平, and Weiwei Ren 任偉瑋. 2015. Songdai hanzhou foiao yu shisu shehui guanxi yanjiu 宋代杭州佛教與世俗社會關係研究. *Ningxia Daxue Xuebao* 寧夏大學學報 5: 119–30.

Zhao, Qingshan. 2013. 5 jian wenshu suo fanyin de dunhuang tufan shiqi xiejing huodong 5件文所反映的敦煌吐蕃期活. *Zhongguo Zangxue*中國藏學 4: 99–104.

Zheng, Binglin 鄭炳林. 2004a. Wantang wudai Dunhuang diqu renkou bianhua yanjiu 晚唐五代敦煌地人口化研究. *Jiangxi Shehui Kexue* 江西社會科學 12: 20–30.

Zheng, Binglin. 2001. Beijing tushu guan cang "wu heshang jinglun mulu" youguan wenti yanjiu 北京圖書館藏《和尚經綸目錄》有關問題研究. In *Dunhuang xue yu zhongguo shi yanjiu lunji—Jinian sun xiusheng xiansheng shishi yi zhounian* 敦煌學與中國史研究論集—紀念孫修身先生逝世一周年. Lanzhou: Gansu Wenhua Chubanshe.

Zheng, Xianwen 鄭顯文. 2004b. Tangdai sengdao ge jiqi fuyuan zhi yanjiu 唐代《僧道格》及其復原之研究. *Pumen Xuebao* 普門學報 20: 1–30.

Zhou, Qi 周奇. 2008. Tangdai guojia dui sengni de guanli—Yi sengni jizhang yu renou kongzhi wei zhongxin 唐代國家對僧尼的管理—以僧尼籍帳與人口控制為中心. *Zhongguo Shehui Jingji Shi Yanjiu*中國社會經濟史研究 3: 8–19.