# Peer review of "A Study on the Literacy Rate of Buddhist Monks in Dunhuang during the Late Tang, Five Dynasties, and Early Song Period"

_religions, doi:10.3390/rel13100992_

Round 1

Reviewer 1 Report

Please see my comments attached.

Author Response

I fully accept these comments. Also, I would like to make the point that many of the spelling errors may have been made during the conversion of word to PDF.

Reviewer 2 Report

The article is based on sound scholarship. The author convincingly attempts to determine monastic literacy in Dunhuang. It is highly original, and the impact of this research will be important.

However, it would be best to have the article proofread. There are quite many English mistakes, although the text is generally comprehensible.

Very often a space between words is lacking, but maybe that is a computer problem.

Some other remarks:

-     In the abstract, as well as every now and then in the main text, it might be good to add more dates, so that also readers with less ready knowledge of Chinese history can follow more easily.

-        Line 59: it might be good to explain mahāyāna altar.

-        Lines 150 ff: I do not understand the connection with the research done by Furet and Ozouf. Is this in a similar context?

-    Line 214 and others: what is meant by bishop? This is a Christian term I would personally avoid.

-         Line 284: should it not be Daxing?

-         Line 351: what is huaya?

-        Line 404 and others: it is hard to grasp the link with S.2729. This needs to be better explained. (see certainly line 419: Fada?)

-          Line 500 and others : it should be Mahāprajñāpāramitā

-          Line 726: why Hoa-San?

-       Conclusion: it is well written, but although the hypothesis that the literacy rate also goes for Central China has been advanced with caution, I am not fully convinced (on this point). What about relatively small monastic places, where less education is available, for instance? 

Author Response

Point 1:

In the abstract, as well as every now and then in the main text, it might be good to add more dates, so that also readers with less ready knowledge of Chinese history can follow more easily.

Response 1:

I have add some dates where important people and things appear.

Point 2: Line 59: it might be good to explain mahāyāna altar.

Response 2: I have added some explanations.

Hao (1998) has observed that, like the Central China, where monks were not ordained annually, the “mahāyāna altars” (fangdeng jietan 方等戒坛) set up by the monks in Dunhuang to grant novice monks upasampada were also not set up every year, but only once every few years or even decades.

Point 3: Lines 150 ff: I do not understand the connection with the research done by Furet and Ozouf. Is this in a similar context?

Response 3: 这里引用Furet and Ozouf的研究主要是因为他们使用的“marriage registers”和本文使用的“the signature list of the monks”一样,都是属于“the signature list”

Furet and Ozouf’s study is cited here mainly because their use of “marriage registers” is the same as the use of “the signature list of the monks” in this paper, both being part of “the signature list”

Point 4: Line 214 and others: what is meant by bishop? This is a Christian term I would personally avoid.

Response 4: I have used “the head monk” instead of the “bishop”.

Point 5: Line 284: should it not be Daxing?

Response 5: yes, it is my mistake, and I have changed “Shancai” to “Daxing”

Point 6: Line 351: what is huaya?

Response 6: I have added some explanations.

huaya 畫押 a kind of signature symbol instead of formal characters

Point 7: Line 404 and others: it is hard to grasp the link with S.2729. This needs to be better explained. (see certainly line 419: Fada?)

Response 7: For example, the fact that “Fada” is a monk appeared in S. 2729. As S. 2729 is very famous in Dunhuang studies, I did not make much explanation when I cited it.

Point 8: Line 500 and others : it should be Mahāprajñāpāramitā

Response 8: yes, it is my mistake, and I have corrected it.

Point 9: Line 726: why Hoa-San?

Response 9: The reason I translated 摩訶衍 as Mahayana hoa-san is because he was called “Mahayana hoa-san”.

Point 10: Conclusion: it is well written, but although the hypothesis that the literacy rate also goes for Central China has been advanced with caution, I am not fully convinced (on this point). What about relatively small monastic places, where less education is available, for instance?

Response 10: This is a fascinating and valuable question, but I have to admit that I still have no direct evidence to answer it, and I hope to further study this issue in depth in the future
